# A SEED-AUGMENT-TRAIN FRAMEWORK FOR UNIVERSAL DIGIT CLASSIFICATION

**Vinay Uday Prabhu, Sanghyun Han, Dian Ang Yap, Mihail Douhaniaris & Preethi Seshadri**
UnifyID
Redwood City, CA 94063, USA
{vinay,sang,dyap,mdouhaniaris,preethi}@unify.id

## ABSTRACT

In this paper, we propose a Seed-Augment-Train/Transfer (SAT) framework that contains a synthetic seed image dataset generation procedure for languages with different numeral systems using freely available open font file datasets. This seed dataset of images is then augmented to create a purely synthetic training dataset, which is in turn used to train a deep neural network and test on held-out real world handwritten digits dataset spanning five Indic scripts, Kannada, Tamil, Gujarati, Malayalam, and Devanagari. We showcase the efficacy of this approach both qualitatively, by training a Boundary-seeking GAN (BGAN) that generates realistic digit images in the five languages, and also qualitatively by testing a CNN trained on the synthetic data on the real-world datasets. This establishes not only an interesting nexus between the font-datasets-world and transfer learning but also provides a recipe for universal-digit classification in any script.

## 1 INTRODUCTION

Transfer learning from the *synthetic realm* to the real-world has elicited a lot of attention in the machine learning community recently (See [1; 2; 3]), which typically entails three steps: generating large volumes of synthetic training data (which is often a relatively *cheap* process), synthesizing it using a domain specific recipe to address the *reality gap* [4], and training real-world deployment-worthy machine learning models. As seen in [1; 2; 3], deep generative models such as Variational Autoencoders (VAE) and Generative Adversarial Networks (GAN) are often deployed during the synthesizing process. This paper fits squarely into this category of work, where we tackle the problem of absence of MNIST-scale datasets for Indic scripts to achieve high, real-world accuracy digit classification by using synthetic datasets generated by harnessing the Open Font License [1] (OFL) font files freely available on the internet.

The main contributions of this paper are:

1. New handwritten, MNIST-styled *Indic-digits* datasets for the following five languages: Kannada, Tamil, Gujarati, Malayalam and Devanagari with 1280 images each.

2. The Seed-Augment-Transfer (SAT) framework for real-world digit classification. This framework can be extended in a myriad of ways to potentially cover all digit representations in various language scripts.

3. Successfully trained digit-GAN models for the Indic languages listed above.

4. Open-sourced scripts and code[2] to reproduce the results disseminated in this paper.

The rest of the paper is organized as follows. In Section 2, we cover in detail the real-world dataset generation process. In Section 3, we introduce the SAT framework. In Section 4 we present the classification and the generative model results, and conclude the paper and discuss extended work in Section 5.

---

[1] https://en.wikipedia.org/wiki/SIL_Open_Font_License
[2] https://anonymous.4open.science/repository/3246a9f1-f5fa-4a74-9822-0dd34b3f8b16/

## 2    DATASET PREPARATION

We begin with a handwritten grid of 32 x 40 digits on a commercial Mead Cambridge Quad Writing Pad, 8-1/2" x 11", Quad Ruled, White, 80 Sheets/Pad book with a black ink Z-Grip Series — Zebra Pen. We then scan the sheet(s) using a Dell - S3845cdn scanner[3] and use an image segmentation script to slice the scanned image grid to generate 1280 28 x 28 *mnist-ized* digit images. An example of the raw scanned image for the *Devanagri-Hindi* script is as shown in Fig 1, with the class-wise means of the handwritten digit images for the five languages are as shown in Fig 2.

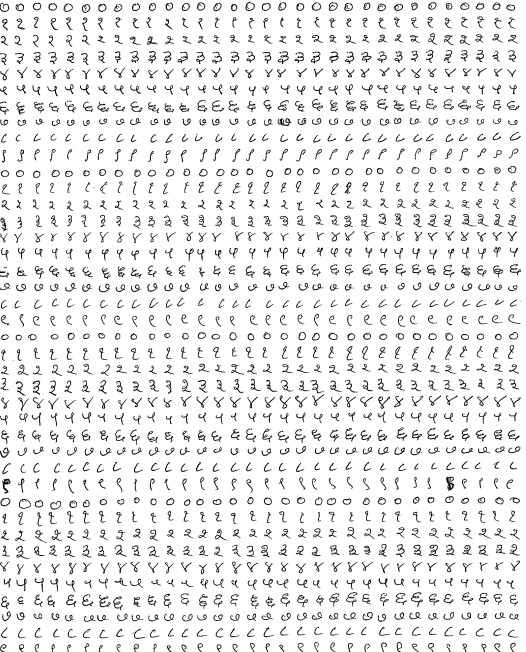

Figure 1: The raw scanned image for the Devanagari handwritten digits dataset

We now perform a novel 2-step sanity check for MNIST compatibility for the languages in the following manner.

### 2.1    COMMONALITY OF THE ZERO DIGIT

Firstly, we pick images belonging to class 0 and pass it through a CNN pre-trained on the MNIST dataset [5]. Given that the representation of 0 is the same in all the Indic languages and the *standard-MNIST*, we expect to get high accuracy predictions for this class.

### 2.2    MORPHOLOGICAL SIMILARITY OF CERTAIN DIGITS WITH REGARDS TO THE STANDARD-MNIST DIGITS

Given the Indian roots of the modern Hindu-Arabic numeral system [6], we expect to find morphological similarity in the shape of certain digits between the modern digit system (used in MNIST) and the Indic languages considered in this work. We exploit this similarity to perform our second sanity check, which is showcased with a specific example in Fig 2, where the shapes of the digits for 3 and 7 in Kannada look similar to 2 in the modern script for the Hindu-Arabic numeral system.

Therefore, passing images belonging to class 3 and class 7 in the Kannada dataset through the standard MNIST-trained CNN should result in high rates of *correct* misclassifications as class 2. For this dataset, we obtained accuracies of $1.0$ and $(1.0, 0.977)$, respectively, for the two checks above.

---

[3]https://goo.gl/fykK8k

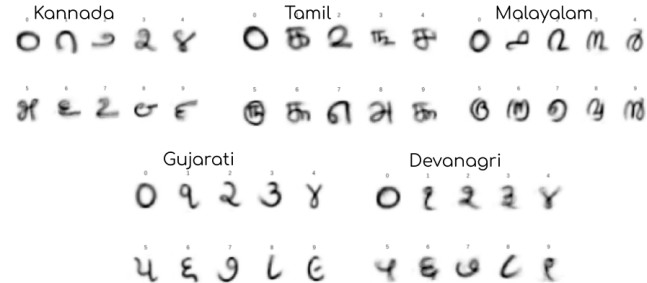

Figure 2: Mean of the 0-9 digits for the languages in the written dataset

# 3 THE SEED-AUGMENT-TRAIN/TRANSFER FRAMEWORK

In the sub-sections below, we present the Seed-Augment-Train/Transfer framework (See Fig 3).

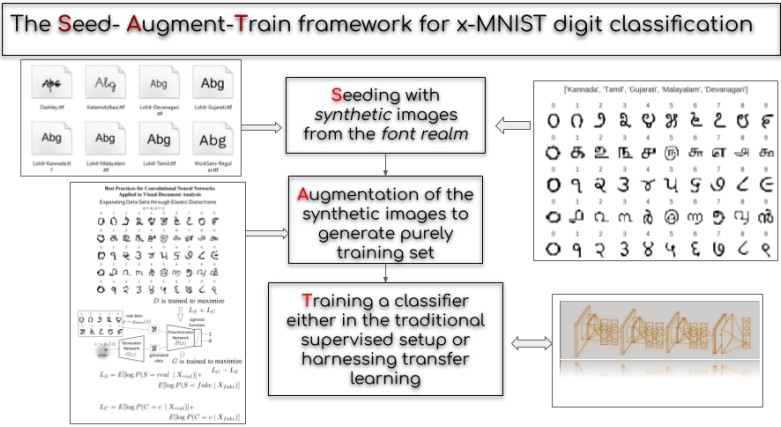

Figure 3: The Seed-Augment-Train/Transfer framework

## 3.1 SEEDING WITH TRUE-FONT IMAGES

There exists a vast body of handwritten as well as synthetic and artistically inscribed textual data freely available on the internet in the form of *font files*. We found 961 Open Font License (OFL) fonts in the *Google fonts* repository [4] alone, each with intra-font variants such as italicized, bold, etc. For the Indic scripts, Red Hat [5] released the *Lohit* [6] font family that covers 11 languages: Assamese, Bengali, Gujarati, Hindi, Kannada, Malayalam, Marathi, Oriya, Punjabi, Tamil, Telugu. In this paper, we used the Lohit True Type Font (TTF)[7] files (that can be used on both Mac and Windows platforms) to create a *seed dataset* for each of the languages.

We combine the text insertion ability of the Python Imaging Library [7] with the textual glyphs generated using the Lohit font family, as per Appendix Listing 1. Using this script, we have a $5 \times 10$

---

[4]https://github.com/google/fonts/tree/master/ofl
[5]https://en.wikipedia.org/wiki/Red_Hat
[6]https://en.wikipedia.org/wiki/Lohit_fonts
[7]https://scripts.sil.org/cms/scripts/page.php?site_id=nrsi&id=iws-chapter08

array of the $28 \times 28$ images as shown in Fig 4. We can similarly create a plethora of seed image datasets using a vast array of other fonts from sources such as Google fonts[8] and Baraha [9].

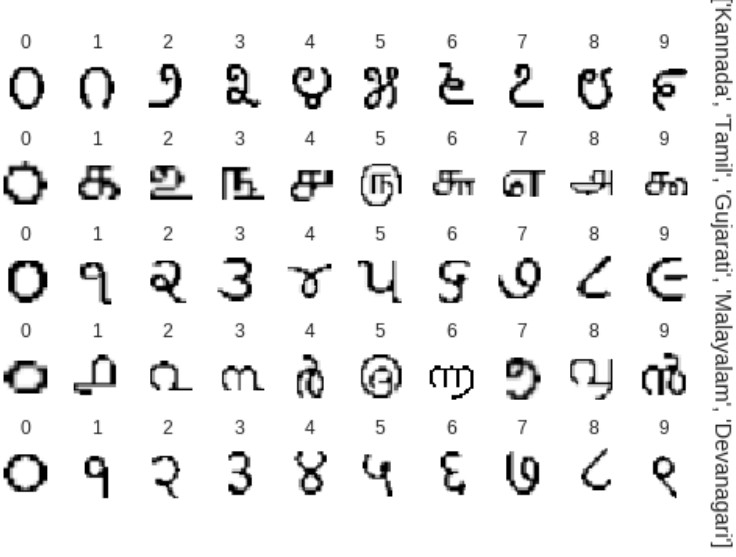

Figure 4: The Lohit seed dataset portraying 0-9 digits for the five Indic scripts

## 3.2 AUGMENTATION

One can argue that the glyph in a seed image is a proper representation of what the digit should look like. However, if our goal is to perform human handwritten digit classification, we need to intelligently distort seed images in order to generate a synthetic dataset *realistic* enough to transfer well into the real-world data domain. The problem of *realistic* augmentation for digits has a variety of approaches. One of the most notable works is Simard et al. [8], which introduces the *Elastic distortion* method hypothesized to correspond well to *"uncontrolled oscillations of the hand muscles, dampened by inertia."* This constitutes our shallow augmentation approach. Once we generate the elastic-distortion augmented dataset, we train an AC-GAN [9] whose samples constitute the *deep augmented* synthetic training dataset.

### 3.2.1 SHALLOW AUGMENTATION

Classic image augmentation techniques [8] include additive noise, intensity shifting and scaling, random rotation, translation, flipping, and warping. As discussed above, we will focus on the Elastic distortion method proposed in [8]. It is a bi-parametric method with the following control parameters:

- $\alpha$: The random-field parameter
- $\sigma$: The Gaussian-convolution standard deviation parameter

As recommended in [8], we set $\alpha = 8$ for each of the five Indic scripts and vary the $\sigma$ parameter and tune until the perturbed images look *realistic*. For these experiments, we used the implementation in *imgaug* [10] python library.

Fig 5 shows a grid of random images generated using the five Indic seed datasets, with $\alpha = 8$ and $\sigma$ varying from $0.01$ to $3$. The realistic-looking image distortions mimicking the variations caused by the human hand occur in the interval of $\sigma \in [1.5, 2.5]$. We use this heuristic to generate the Elastic distorted datasets for all the five languages by sampling from $\sigma \sim U[1.5, 2.5]$

---

[8] https://fonts.google.com/
[9] https://www.baraha.com/
[10] https://imgaug.readthedocs.io

Figure 5: The Lohit seed dataset portraying 0-9 digits for the five Indic scripts

### 3.2.2 DEEP AUGMENTATION - ACGAN

We complement classic image augmentation techniques using synthetic data generation, where data is generated and sampled from deep generative models such as Generative Adversarial Networks (GANs). GANs can implicitly learn the true data distribution and generate promising samples such that the samples from the learned distribution resemble the true underlying data distribution $p_{data}$.

In order to generate synthetic data with corresponding labelled classes, we leverage the Auxiliary Classifier GAN (ACGAN) in which the generator input can be conditioned to sample from a target class [9]. ACGAN is trained with the discriminator producing two outputs, one to discriminate between real and fake images $X$, and the other to classify $X$ in terms of its class, $c$.

During training, we explicitly sample $X_c$ from the generator during training for each class $c \sim p_c$, with the generator parameters adjusted to maximize the superposition of the two components in the objective function, $L_s$ and $L_c$ as follows. Here $L_s$ refers to the log-likelihood of the correct source whereas $L_c$ refers to the log-likelihood of the correct class.

$$L_s = \mathbb{E}\left[\log P(S = real \mid X_{real})\right] + \mathbb{E}\left[\log P(S = fake \mid X_{fake})\right] \tag{1}$$

$$L_c = \mathbb{E}\left[\log P(C = c \mid X_{real})\right] + \mathbb{E}\left[\log P(C = c \mid X_{fake})\right] \tag{2}$$

The discriminator, $D$, is trained to maximize $L_s + L_c$ whereas the generator, $G$, is trained to maximize $L_c - L_s$. In our experiments we used learning rate $l$ with Adam Optimizer ($l = 0.0002, \beta_1 = 0.5, \beta_2 = 0.999$), and with dropout probability $p_{dropout} = 0.3$ in the discriminator and batch normalization in the generator as regularization techniques. The generator takes in the Hadamard product between the latent space and the class conditional embedding as an input.

ACGAN exhibits promising results in deep class-condition augmentation, as the discriminator now produces a probability distribution over the sources and also a probability distribution over the class labels. With the introduction of class probabilities, $p_c$, more structure is present in the GAN latent space which helps produce higher-quality training samples and improve stability during training. Nonetheless we chose ACGAN as a key deep generative model in the *augment* step as it generates samples with corresponding class labels, as the class is crucial in measuring classification accuracy for the generated output with regards to the corresponding label. Fig 11 showcases the snapshots of images produced by sampling the ACGAN generator after 10 epochs for all the five languages. For each language we see two blocks distanced by the white separator space. The images in the left *column* are the generated images and the images in the right column are *real* shallow-synthetic images that the GAN was trained on.

## 4 RESULTS

We wish to showcase the efficacy of the SAT framework using two approaches. The first approach is more qualitative by training a deep-generative model (BGAN) using only shallow synthetic data, and utilizing the model to generate realistic looking handwritten digit images for the various languages. The second approach is quantitative where we train a Convolutional Neural Network (CNN) solely on synthetic training data, and test the efficacy with both real-world handwritten data as well as *doping* the training dataset with a small amount of real-world data ($\sim 20\%$). We then computing the classification accuracies in both scenarios.

### 4.0.1 BOUNDARY-SEEKING GAN TRAINING AND SAMPLE GENERATION

GANs face the limitation where the generated samples have to be completely differentiable with regards to the parameters of the generator, $\theta$, and do not work for discrete data where the gradient is zero almost everywhere, not to mention other issues such as training stability. Boundary-seeking GANs (BGANs) allow for the training of GANs with discrete data by using a policy gradient based on the KL-divergence with importance weights as a reward signal [10].

In our setting, we focus more on the convexity of the objective function of BGANs. We formulate BGANs in our setting as follows: we have $G_\theta : \mathcal{Z} \to \mathcal{X}$ as a generator that takes in a latent variable drawn from a prior, $z \sim p_z$. We also have a neural network $\mathcal{F}_\phi : \mathcal{X} \to \mathbb{R}$ parameterized by $\phi$, with our BGAN objective defined as:

$$\hat{\theta} = \arg\min_\theta \mathbb{E}_{z \sim p_z(z)} \left[ \frac{1}{2} \left( \log D(x) - \log(1 - D(x)) \right)^2 \right] \tag{3}$$

Typical generator loss of continuous data is a concave function with poor convergence properties since it relies on continuous optimization of the discriminator to stabilize learning. We reformulate the concave optimization as a convex one by training the generator to aim for the decision boundary of the discriminator, which improves stability. In implementation we also use Adam Optimizer ($l = 0.0002, \beta_1 = 0.5, \beta_2 = 0.999$).

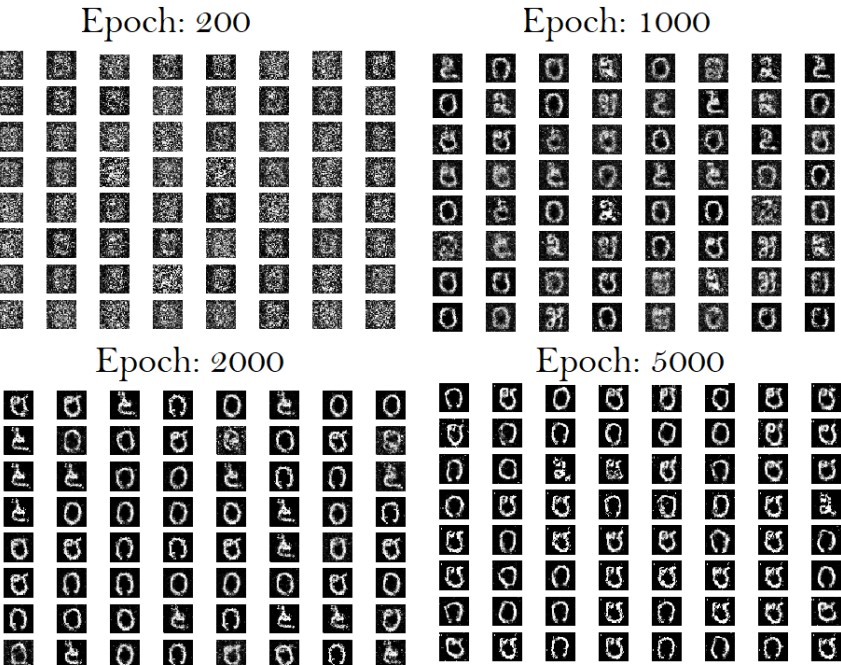

Figure 6: Snapshots of the images produced by sampling the BGAN generator after different epochs

The results are as shown in Fig 6 for epoch numbers 200, 1000, 2000 and 5000. We see the *learning* setting in as the epochs increase with realistic looking imagery emerging after as few as 5000 epochs.

## 4.1 CLASSIFICATION RESULTS

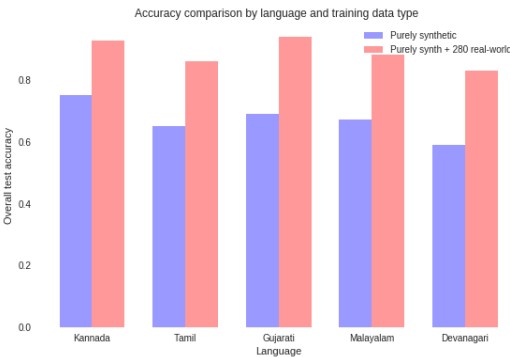

Figure 7: The accuracy comparisons for the five scripts.

For the test-accuracy computations over the real-world dataset we trained an off-the-shelf *plain-vanilla* CNN model[11] with cross-entropy loss and the ada-delta optimizer. Fig 7 showcases the results obtained. When we used a purely synthetic training dataset with 140,000 training samples of which 80,000 came from the elastic distortion procedure from section-3 and 60,000 came from the ACGAN, we obtained overall accuracies varying from $60\%$ to $76\%$ for the five different languages (the `Purely synthetic` bars in the bar-plot). When we augmented this training dataset with merely $\sim 20\%$ of the real-world dataset (a 280-1000 train-test split) the accuracies shot up by a significant amount.

We believe this observation has two important facets to it besides the *reality-gap*: the limited size of the test-set and some idiosyncratic morphological changes in the digits' representation in the lohit-font and the colloquial representation. For example both in Gujarati and Kannada (See Fig 10b and Fig 10a respectively) we see that six is represented rather differently in the synthetic set and in the test set. This seems to get tackled when a small amount of real-world data is introduced into the training data. In the case of Gujarati, the accuracy for digit 6 increases from $0.7\%$ to $95\%$!

As seen in Fig 7 the overall test accuracy increases for every language by a substantial margin for the 'Purely synth + 280 real-world training' dataset option. The confusion matrices for the two training scenarios are as shown in Fig 8 and Fig 9 respectively.

## 5 CONCLUSION AND FUTURE WORK

We introduced 5 new real-world digit datasets in Indic languages and also a transfer learning framework which we term as Seed-Augment-Train/Transfer (SAT) to perform real-world handwritten digit classification in these languages. Our goal is to draw the attention of the machine learning community to this potentially attractive nexus between the work happening in the synthetic-to-real transfer learning domain and a veritably rich reservoir of semi-synthetic textual glyphs training data freely available in the form of *Font files*.

We showcased the efficacy of this SAT-based approach by perform digit-classification and also training GANs for the digits of the languages considered. We are expanding this foundational effort in several directions: the first entails using a small portion of the dataset for transfer learning, which will help tackle cases in which off-the-shelf image augmenters do not effectively capture human handwriting's natural variations. The second involves generative modeling of the ***deviations*** in handwritten digits with regards to the clean seed-synthetic dataset using the extended MNIST dataset (in lieu of the digits themselves). We would then apply this approach to *deep augment* seed datasets

---

[11]https://github.com/keras-team/keras/blob/master/examples/mnist_cnn.py

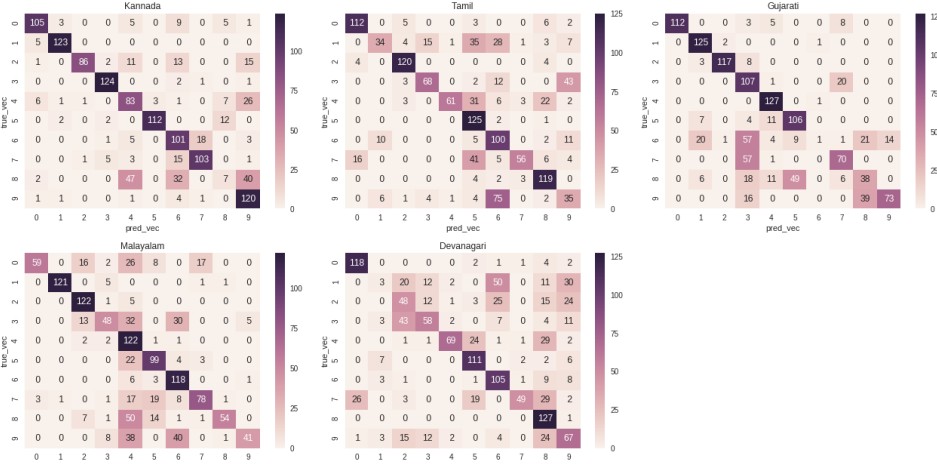

Figure 8: Class-wise confusion matrix for all the five Indic scripts obtained after training on purely synthetic hybrid datasets.

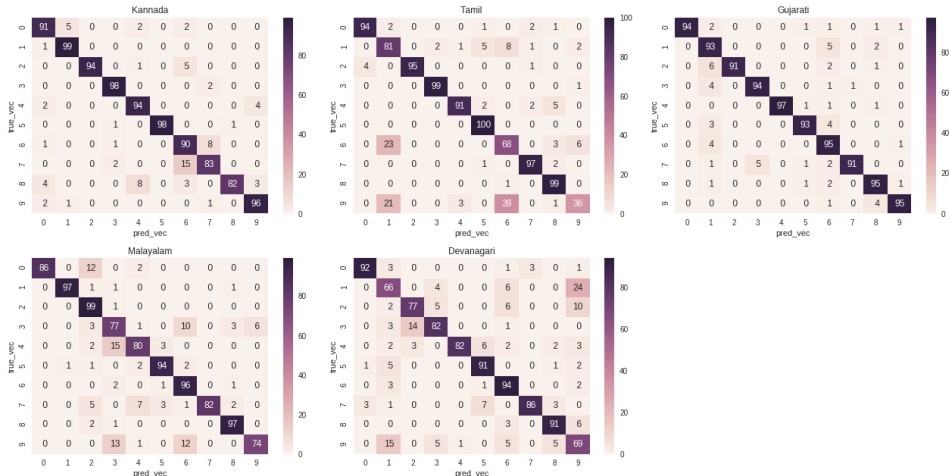

Figure 9: Class-wise confusion matrix for all the five Indic scripts obtained after training on synthetic data doped with 280 training samples from the real-world datasets.

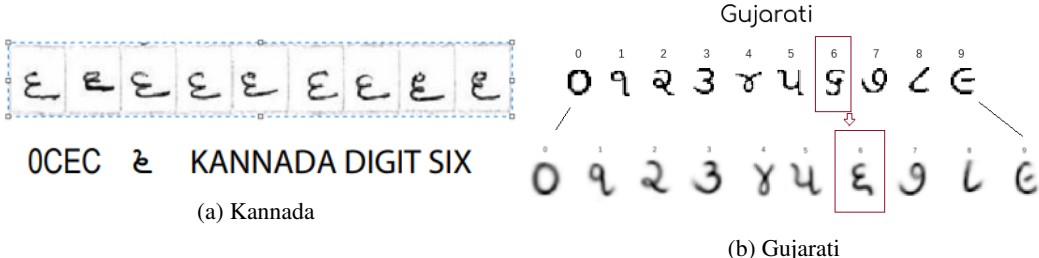

(a) Kannada

(b) Gujarati

Figure 10: Changes between the Lohit-font representation and the vernacular written representation of the digit 6 in *(a) Kannada, (b) Gujarati*

across other languages as well, resulting in a universal-MNIST dataset recipe. Thirdly, we are collecting larger volumes of real world handwritten test data for these languages to form MNIST sized datasets.

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

## 6 APPENDIX

```python
import numpy as np

def num2image(i):
    image = Image.new('L', (28, 28))
    draw = ImageDraw.Draw(image)
    # Use a truetype font
    font = ImageFont.truetype("K.ttf",
    25)
    draw.text((7, 1),i, font=font,
    fill=(255))
    return np.array(image)
```

Listing 1: Python example utilizing Python Imaging Library for text insertion ability.

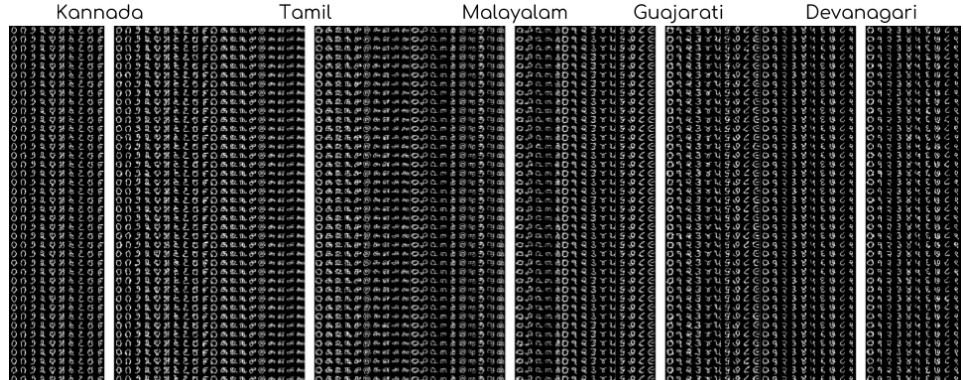

Figure 11: Snapshots of the images produced by sampling the ACGAN generator after 10 epochs for all the five languages.

```
Layer (type)                 Output Shape              Param #
=================================================================
conv2d_1 (Conv2D)            (None, 26, 26, 32)        320
_________________________________________________________________
conv2d_2 (Conv2D)            (None, 24, 24, 64)        18496
_________________________________________________________________
max_pooling2d_1 (MaxPooling2 (None, 12, 12, 64)        0
_________________________________________________________________
dropout_1 (Dropout)          (None, 12, 12, 64)        0
_________________________________________________________________
flatten_1 (Flatten)          (None, 9216)              0
_________________________________________________________________
dense_1 (Dense)              (None, 128)               1179776
_________________________________________________________________
dropout_2 (Dropout)          (None, 128)               0
_________________________________________________________________
dense_2 (Dense)              (None, 10)                1290
=================================================================
Total params: 1,199,882
Trainable params: 1,199,882
Non-trainable params: 0
_________________________________________________________________
```

Figure 12: The plain-vanilla CNN-MNIST architecture

