# OpenReview forum: "A Seed-Augment-Train Framework for Universal Digit Classification "
_ICLR.cc/2019/Workshop/DeepGenStruct — DeepGenStruct 2019_

### Official Review · AnonReviewer2 · 2019-04-15

**Rating:** 3
**Confidence:** 1

**Review:**

This paper presents new datasets for give languages and proposes a new framework (SAT) for font image datasets generation. I think this paper makes reasonable contribution to the literature.

---

### Official Review · AnonReviewer1 · 2019-04-16
**digit data augmentation**

**Rating:** 2
**Confidence:** 2

**Review:**

This work aims to create handwritten digit data like MNIST in other languages. The authors started with open fonts dataset and then applied image augmentation techniques to add distortions. Finally, the authors collected real handwritten digit data, and trained with BGAN to generate more handwritten like images with labels. The authors showed that direct training on the synthetic dataset gets 60-76% accuracy, and adding a small amount of real-world data gets a substantial improvement.

Pros:
1. It's clearly written and easy to follow.
2. The authors showcase a working example of synthetic-to-real transfer learning, which could be interesting to the broader ML community.

Cons:
1. Ablation study missing. What would the results be if we just use the GAN generated part, and what if we only use the rest?

Overall I think this paper is intersting, but I don't consider it to be very relevant for this Deep Generative Models for Highly Structured Data workshop, since it's a direct application of GAN and might be more relevant for OCR venues.

---

### Decision · Program_Chairs · 2019-04-19
**Acceptance Decision**

**Decision:**

Accept

**Comment:**

Although the scores are below our average acceptance rate, we believe this paper has interesting contributions:

1. The set of chosen languages is diverse.
2. Synthetic and real dataset for digits in these languages
3. Interesting way of using GANs to improve the downstream task.